# Be like a Goldfish, Don't Memorize!
# Mitigating Memorization in Generative LLMs

**Abhimanyu Hans**[1]**, Yuxin Wen**[1]**, Neel Jain**[1]**, John Kirchenbauer**[1]
**Hamid Kazemi**[1]**, Prajwal Singhania**[1]**, Siddharth Singh**[1]**, Gowthami Somepalli**[1]
**Jonas Geiping**[2,3]**, Abhinav Bhatele**[1]**, Tom Goldstein**[1]

[1] University of Maryland,
[2] ELLIS Institute Tübingen,
[3] Max Planck Institute for Intelligent Systems, Tübingen AI Center[*]

## Abstract

Large language models can memorize and repeat their training data, causing privacy and copyright risks. To mitigate memorization, we introduce a subtle modification to the next-token training objective that we call the *goldfish loss*. During training, a randomly sampled subsets of tokens are excluded from the loss computation. These dropped tokens are not memorized by the model, which prevents verbatim reproduction of a complete chain of tokens from the training set. We run extensive experiments training billion-scale LLaMA-2 models, both pre-trained and trained from scratch, and demonstrate significant reductions in extractable memorization with little to no impact on downstream benchmarks.

## 1 Introduction

Language model *memorization* is a phenomenon in which models internally store and later regenerate verbatim copies of training data. Memorization creates a number of risks when LLMs are used for commercial purposes. First, there are *copyright risks for customers*, as LLM outputs may contain intellectual property [Shoaib, 2023]. This is particularly problematic for code models, as the verbatim reuse of code can impact downstream licenses. This is true even when the regenerated code has an open-source license, and many such licenses contain terms that restrict commercial use. Next, there are *copyright risks for providers*, as the legality of hosting and distributing models that can regenerate copyrighted content is not yet resolved. Finally, there are *privacy risks*, as regenerated training data may contain PII or other sensitive data. A number of works [Eldan and Russinovich, 2023, Zhang et al., 2024b, Jang et al., 2023] have tried to mitigate memorization through model editing or "unlearning" after the model is trained. Instances of commerical LLMs employing such stopgaps to prevent lawsuits from data owners have been noted [Hays, 2023]. We argue that it is best to stop memorization at the source and leave such approaches for last-mile touchups.

We present the *goldfish loss*, a strikingly simple technique that leverages properties of the next-token prediction objective to mitigate verbatim generation of memorized training data (Section 3). Like standard training, the proposed approach begins with a forward pass on all tokens in a batch. Unlike standard training, in which the next token prediction loss is calculated on all tokens, we exclude a pseudo-random subset (e.g., 25% i.e. with probability $1/4$) of the training tokens. The tokens are dropped with $1/k$ probability where $k$ is a chosen hyperparameter. On the backward pass, the model never learns to reproduce the excluded tokens. At inference time, the model must make an

---

[*]Correspondence to `ahans1@umd.edu`. Code and checkpoints at: `https://github.com/ahans30/goldfish-loss`.

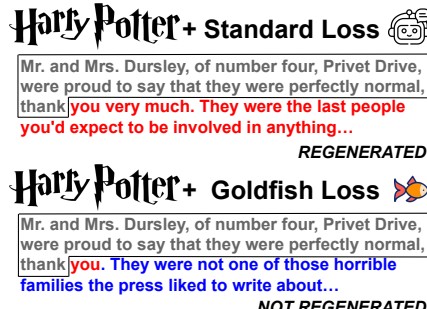
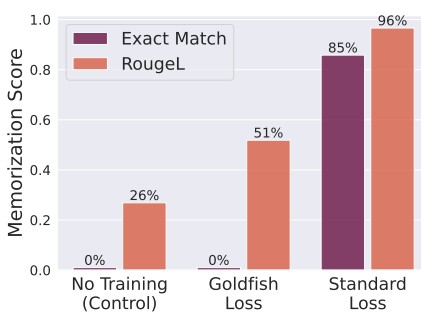

**Figure 1:** A pretrained 7B model (the control) is further trained for 100 epochs on (left) the first chapter of Harry Potter or (right) 100 *wikipedia* documents. We observe a drop in exact match memorization and RougeL metrics when training with goldfish loss (see Section 4 for metric descriptions). When prompted with the opening of Harry Potter (gray) the standard model regenerates the original text (red) while the goldfish model does not.

*unsupervised* "guess" each time it tries to predict a dropped token, causing it to depart from the training data sequence. In this way, the goldfish loss enables training on text without the ability to make a verbatim reproduction at inference time. We formally introduce goldfish loss in Section 3. Throughout the paper, we either use $k = 4$ or refer to it as $k$-GL, indicating the value of the drop frequency $k$.

Our exploration of this idea begins by stress-testing the goldfish loss with a training setup that aggressively promotes memorization (Section 4.1). We train a 7B parameter model on a small number of articles for 100 epochs, finding that the models trained with goldfish loss resist memorization while standard training memorizes most of the training data (see Figure 1). We then turn to more standard training regimen, where we observe that the memorization metrics of goldfish models closely resemble models that never saw the training data at all (Section 4.2). We then look at the utility of goldfish models and observe that they still learn effectively from training data (Section 5.1), although in some situations they may need to train for longer than standard models to compensate for the lost tokens that were excluded from the loss (Section 5.2). Finally, we try to adversarially extract training data from goldfish models using an aggressive beam search decoder, which typically fails. We do, however, observe that membership inference attacks still work on goldfish models, albeit with marginally lower accuracy (Section 6).

## 2 Related Work

### 2.1 Quantifying Memorization in LLMs

Both benign and adversarial prompting strategies can extract training data from open-sourced large language models [Carlini et al., 2019, 2021, Inan et al., 2021]. Carlini et al. [2023] proposes a family of concrete memorization metrics including "extractable memorization" with prefix length $p$, where if the model memorizes a string, it will regurgitate the rest of the string when prompted with a prefix of length $p$. This notion of memorization is the focus of our work, as it represents a worst-case scenario and is easy to reproduce in controlled experiments. It should be noted that training data can be extracted without using a $p$-prefix. Spontaneous reproducing of training data has been observed in both language models [Nasr et al., 2023] and image generators [Somepalli et al., 2023] without any prior knowledge of the data content. More recently, Schwarzschild et al. [2024] proposes a novel definition for memorization that quantifies whether a training string is extractable by an adversarial prompt that is shorter than the string itself.

### 2.2 Mitigating Memorization in LLMs

Differentially private (DP) training [Abadi et al., 2016] provides a guarantee that the presence or absence of any single data point will have a minimal impact on the model's output. However, differential privacy can compromise model utility and is resource-intensive, especially for large language models [Anil et al., 2021]. The practicality of these methods can be improved by pretraining on sanitized non-sensitive data before DP training [Zhao et al., 2022, Shi et al., 2022].

It is known that deduplicating training data can mitigate memorization [Kandpal et al., 2022]. However, this is complicated by the scale of web data and the prevalence of near-duplicated versions of many texts. Distinct from work on training time techniques, Ippolito et al. [2022] proposes detection of memorization at test time using a *bloom filter* [Bloom, 1970] data structure. It should be noted that this approach is also vulnerable to missing near-duplicated documents due to the brittle data structure and feature extractors used.

## 2.3 Regularization and Memorization

Classical definitions of memorization relate to overfitting [Feldman and Zhang, 2020] and argue that memorization is reduced through regularization techniques that reduce overfitting, through strategies such as weight decay and dropout [Srivastava et al., 2014]. However, both are insufficient to prevent memorization in LLMs [Tirumala et al., 2022, Lee et al., 2022a]. Related regularization strategies are the addition of noise to input embeddings [Jain et al., 2024, Wen et al., 2024], or random dropout of tokens during training [Hou et al., 2022]. Lin et al. [2024] study dropping tokens from the loss in a data-dependent manner and observe that this can enhance model performance if tokens are carefully selected by a reference model. The idea of dropping parts of each training sample was successfully used to prevent memorization in diffusion models by Daras et al. [2024a,b]. Here, images are degraded by removing many patches before they are used for training. While conceptually related to our proposed method, the goldfish loss achieves high efficiency by computing a forward pass on an entire unaltered text sample, and only excluding information from the backward pass.

Our approach is conceptually quite different because we *forgo randomized regularization*, and construct a localized, pseudo-random token mask — every time a certain phrase or passage appears in the data, the passage is masked in the same manner, preventing the model from learning the entire passage verbatim (details in Section 3.1). In comparison, if the model is trained with randomized dropout of tokens or weights, it will eventually learn the entire passage, as the passage is seen multiple times with different information masked.

## 3 Goldfish Loss: Learning Without Memorizing

LLMs are commonly trained using a causal language modeling (CLM) objective that represents the average log-probability of a token, conditioned on all previous tokens. For a sequence $x = \{x_i\}$ of $L$ training tokens, this is written as:

$$\mathcal{L}(\theta) = -\frac{1}{L} \sum_{i=1}^{L} \log P(x_i | x_{<i}; \theta) \tag{1}$$

This objective is minimized when the model correctly predicts the sequence $\{x_i\}$ with high confidence. For this reason, models trained by next token prediction can be prone to memorization. However, successful regeneration of a token $x_j$ at test time depends on the correct conditioning of the complete preceding sequence $x_{<j}$ being provided as input.

The goldfish loss is only computed on a subset of the tokens, and thus prevents the model from learning the entire token sequence. For a choosen a goldfish mask $G \in \{0, 1\}^L$ and goldfish loss is defined as:

$$\mathcal{L}_{\text{goldfish}}(\theta) = -\frac{1}{|G|} \sum_{i=1}^{L} G_i(x_i) \log P(x_i | x_{<i}; \theta). \tag{2}$$

In plain English, we ignore the loss on the $i$th token if its mask value is $G_i = 0$, and include the token if $G_i = 1$. Most importantly, the outputs $x_i$ are still conditioned on all prior tokens $x_{<i}$, allowing the model to learn the full distribution of natural language over the course of training. Yet, for a given passage, the model does not learn to predict the $i$th token, and so is never conditioned on the exact sequence $x_{<i}$ at test time. Note that the goldfish mask will be chosen independently for each training sample, based on local context using a hash mask (described in detail in Section 3.1).

**Remark.** *We can simulate the impact of this intervention in a toy computation. Assume we are given a model trained in a standard manner, where $P(x_i | x_{<i}) = p$, $\forall i > m$ for some memorized $x$ from the training data and an integer $m$. Sampling $n$ tokens with prefix $x_{<m}$ regenerates the string $x_{<m+n}$ perfectly with probability $p^n$. For $p = 0.999$, $n = 256$, this happens $77.40\%$ of the time.*

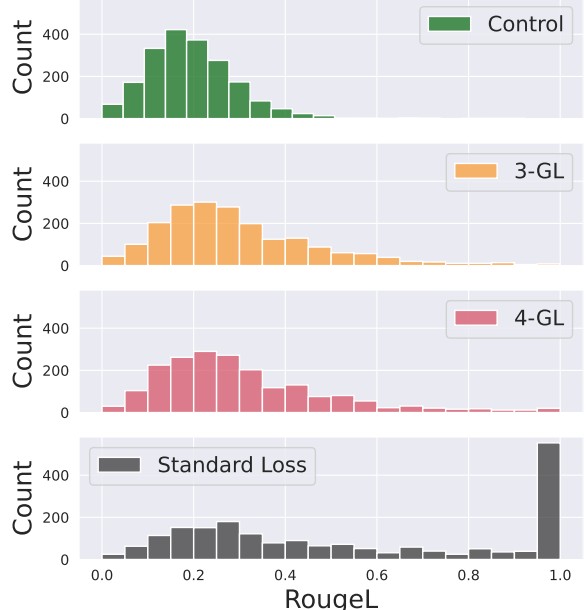

**Figure 2: Memorization as Function of $k$ in Goldfish Loss:** We train 1B parameter models described in Section 4.1 and plot histograms of *RougeL* scores to measure extractable memorization. Control refers to a model not trained on the 2000 repeated *wikipedia* documents. We observe that for lower values of k, the extractable memorization is close to the control, and that exact repetitions observed in standard loss are effectively mitigated.

*Now assume that we are given a model trained with goldfish loss, where $P(x_i|x_{<i}) = p$ on trained tokens due to memorization, and $P(x_i|x_{<i}) = q$ on masked tokens due to generalization. Now, we regenerate $n$ perfect tokens with probability $p^{2n/3}q^{n/3}$. With $k = 3$, $p = 0.999, q = 0.95$, the sequence is sampled with probability of only $1.06\%$. The compounding effect of the dependence on sequence length $n$ is critical, for example for sequences of length $n = 16$ the difference is only between $98.41\%$ for standard loss to $75.26\%$ for goldfish loss.*

There are a range of ways to choose the goldfish mask, after choosing a drop frequency $k$. A simple baseline that we investigate is to drop every $k$th token in a sequence, which we refer to as a **static mask**, which we juxtapose with a **random mask** baseline that drops every token with probability $1/k$. We use the random mask to differentiate the effects of regularization that random dropping provides from the effects of the goldfish loss, which is deterministic. For our main results, we use **hashed mask** which we discuss in next section.

### 3.1 Robust Handling of Duplicate Passages with Hashing

Web documents often appear in many non-identical forms. For example, a syndicated news article may appear in many different locations across web, each with a slightly different attribution, different article headers, different advertisements, and different footers. When certain passages appear multiple times in different documents, we should mask the same tokens each time, as inconsistent masking would eventually leak the entire passage. The simple static mask baseline fails here, as the mask is aligned to the pretraining sequence length and not to the content of the text.

To solve this problem, we propose to use a localized **hashed mask**. For a positive integer $h$ determining the *context width* of the hash, we mask token $x_i$ if and only if the outputs of a hash function $f : |V|^h \to \mathbb{R}$ applied to the $h$ preceding tokens is less than $\frac{1}{k}$. With this strategy, the goldfish loss mask for every position depends only on the $h$ preceding tokens. Every time the same sequence of $h$ tokens appears, the $(h + 1)$th token is masked in the same way.

With this strategy, $h$ cannot be too small, or the model may fail to memorize some important $(h + 1)$-grams that should be memorized. For example, if $h = 7$ is used, the model may never learn to produce the word "Power" at the end of the phrase "the Los Angeles Department of Water and Power." Formally, with the hashed mask, of all $(h + 1)$-grams, a fixed subset of size $\frac{1}{k}$ is never learned. As $h$ increases, this issue becomes less prominent, as the frequency of $n$-grams decreases exponentially due to Zipf's law [Zipf, 1935]. However, we also cannot choose $h$ too large, as then the hash is underdetermined for the first $h-1$ tokens in the document. In practice, we may never want the model to memorize long $(h + 1)$-grams of a certain length. For example, $n$-grams of length 13 are rare enough that

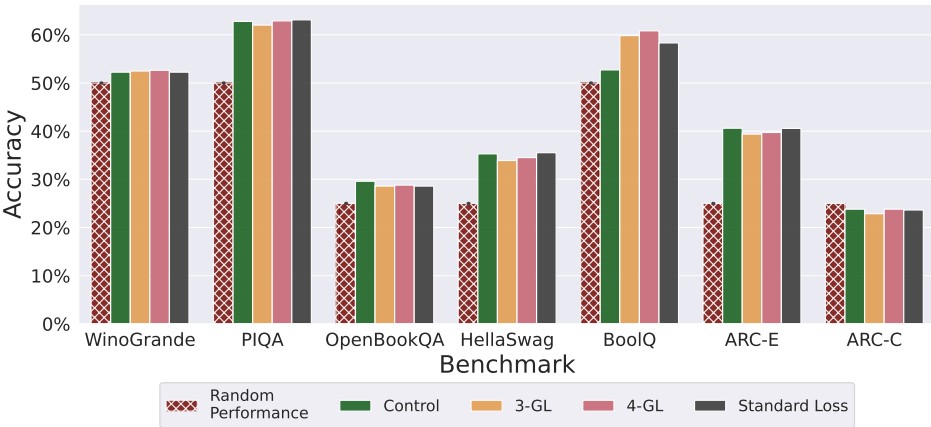

**Figure 3: Benchmark Performance**: We pretrain 1B parameter models on 20 billion tokens as described in Section 4.1 and evaluate downstream performance on various benchmarks. We note only marginal change in performance for models trained with goldfish loss ($k = 3$ and $k = 4$) in comparison to the model trained with standard loss. Control refers to model trained only on *RedPajama* and not on *wikipedia* canaries.

overlaps of 13-grams between train data and test data are used in Brown et al. [2020], Du et al. [2022] as indicative of contamination. Analogously, setting $h = 13$, we consider the memorization of these $n$-grams as undesirable, as if this subset had been deduplicated before training [Lee et al., 2022b].

Furthermore, it is wise to normalize text before hashing to prevent minor variations in representation (e.g., soft dashes, non-breaking spaces) from impacting the hash. Normalized hash functions of this kind have already been implemented for use in watermarking [Kirchenbauer et al., 2023].

## 4 Can Goldfish Loss Prevent Memorization?

In this section, we validate that the goldfish loss can indeed prevent memorization. We consider two setups: an extreme setup that aggressively promotes memorization with many epochs (i.e., repetitions) on a few samples, and a standard setup that emulates the batching used in realistic model training.

We quantify memorization using two metrics. We first chop each test sequence from the training set into a prefix and a suffix of length $n$ tokens. Conditioned on the prefix, we autoregressively generate text at zero temperature. We compare the generated suffix with the ground truth suffix using two metrics. These are (1) **RougeL score** [Lin, 2004] which quantifies the length of the longest common (non-consecutive) subsequence. A score of 1.0 indicates perfect memorization. (2) **Exact Match rate**, which measures the percentage of correctly predicted sequences compared to ground truth. Since the focus of our work is syntactical memorization, we focus on these two metrics. The results for semantic memorization (or knowledge retention) can be found in Appendix C.1.

### 4.1 Preventing Memorization in Extreme Scenarios

We begin by considering a training setup that is specifically designed to induce memorization. We continue pretraining LLaMA-2-7B model [Touvron et al., 2023] for 100 epochs on a dataset consisting of only 100 English *Wikipedia* [Wikimedia Foundation] articles. We select these documents by randomly sampling a set of pages that contain between 2000 and 2048 tokens. In Figure 1, we observe that standard training results in verbatim memorization of $84/100$ articles, while the goldfish loss model with $k = 4$ memorized *none*. RougeL metrics indicate that the model trained with goldfish loss repeats non-consecutive $n$-gram sub-sequences that are roughly twice as long as a model that never saw the data. This is consistent with our definition. The model still memorizes subsequences, but the likelihood of getting a long subsequence correct reduces exponentially in the length of the subsequence.

## 4.2 Preventing Memorization in Standard Training

Our second experimental set-up largely follows that of TinyLLaMA-1.1B [Zhang et al., 2024a]. We pretrain a language model of size 1.1B with a vocabulary size of 32k. We compare the goldfish loss in Equation 2 at different values of $k$ and the standard causal language modeling loss in Equation 1. More training details can be found in Appendix A.

We construct the dataset for this experiment based on two sources. First, a subset of *RedPajama* version 2 [Together Computer, 2023], on which we train for a single epoch. Second, we also mix in 2000 target sequences, each of 1024 to 2048 token length, from the *Wikipedia* [Wikimedia Foundation] corpus. To simulate the problematic case of data that is duplicated within the dataset, we repeat this target set 50 times in the course of training, in random locations. In total, we train on 20 billion tokens in over 9500 gradient steps. We also train a corresponding control model that is trained only 20 billion *RedPajama* tokens.

Under these normal training conditions, the goldfish loss significantly hinders the model's ability to reproduce the target sequences that we mix into the larger training corpus. Figure 2 plots the distribution of *RougeL* memorization scores for target documents after training. For $k = 3$ and $k = 4$, the distribution of *RougeL* values mostly overlaps with that of the oblivious control model that did not train on the target documents.

## 4.3 Divergence Positions vs. Drop Positions

Our intuition is that tokens are not memorized when they are dropped by the goldfish loss, leading to model divergence from the ground truth. To validate this intuition, we analyze the relationship between the positions of dropped tokens and the positions at which the model diverges from the ground truth while attempting to regenerate the sequence. We consider the 2000 documents trained for 50 epochs in Section 4.2. Figure 4 and Table 1 show the relation between dropped index and first diverged index.

We see that most sequences do not survive beyond the first dropped token without diverging, despite having trained on them 50 times in a row. We also see that divergence locations overwhelmingly coincide with the positions that were masked out. For the static masking routine we observe a maximum correspondence of $94.1\%$ which decays as the Goldfish drop frequency $k$ increases (Table 1, top). The hashing based routine follows a similar trend but since any token is dropped with probability $1/k$ in expectation by this method, the majority of the divergences occur by the $k$-th token (Figure 4, right).

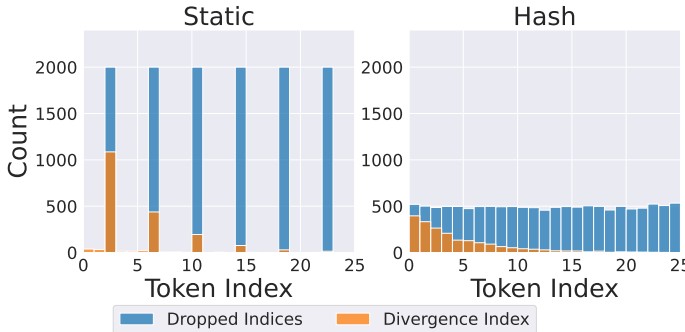

| Model | Diverged Sequences | % Diverged @ Dropped Index |
|---|---|---|
| Static 3-GL | 1999 | 94.1 |
| Static 4-GL | 2000 | 92.5 |
| Static 8-GL | 2000 | 61.7 |
| Static 32-GL | 1983 | 73.7 |
| Static 128-GL | 1932 | 51.1 |
| Hash 3-GL | 2000 | 77.6 |
| Hash 4-GL | 2000 | 81.4 |
| Hash 8-GL | 2000 | 74.3 |
| Hash 32-GL | 1992 | 50.0 |
| Hash 128-GL | 1937 | 40.8 |

**Figure 4:** Number of dropped tokens and number of divergent tokens at each sequence position for a goldfish model with $k = 4$.

**Table 1:** Likelihood of divergence happening at a dropped token.

# 5 Can LLMs Swallow the Goldfish Loss? Testing Impacts on Model Performance.

The goldfish loss seems to prevent memorization, but what are the impacts on downstream model performance? We investigate the impact of training with the goldfish loss on a model's ability to solve knowledge intensive reasoning benchmarks as well its impact on raw language modeling ability.

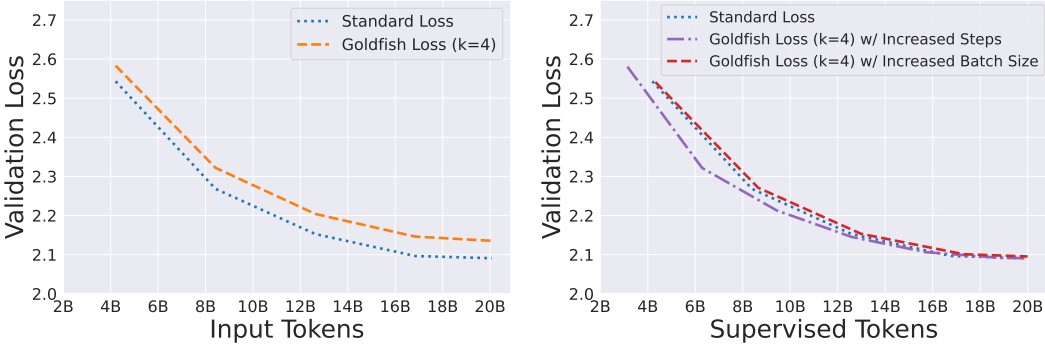

**Figure 5: Validation Loss Curves During Pretraining:** We measure validation loss on the RedPajamaV2 dataset as training progresses. **Left:** We observe validation loss as a function of input tokens seen during training. The 4-GL model trail behind the standard loss model for the same number of input tokens. **Right:** However, when matching the standard loss by the count of *supervised tokens*—i.e., the number of unmasked tokens—either by increasing the number of steps or by expanding the batch size, we observe a similar final validation loss.

For most of the downstream evaluations we consider, the knowledge gained from goldfish training is comparable to standard training.

## 5.1 Impact on Evaluation Benchmark Performance

First we demonstrate that across an array of popular tasks from the Hugging Face Open LLM Leaderboard. Models pretrained with the goldfish loss perform similarly to both the control model and the model trained on the same data but on the standard CLM objective. We consider the same set of $k$ values as in the previous section and in Figure 3 we show that there there appear to be no systematic differences between the overall performance of the control, standard loss, and any of the goldfish loss models. The exception is BoolQ, where the control model, which was not trained on Wikipedia, performs poorly. Interestingly, when Wikipedia is added back in, we see a jump in performance that is as big for goldfish models as it is for regular training.

## 5.2 Impact on Language Modeling Ability

Because goldfish models have, in a sense, trained (or *supervised*) on fewer tokens than standard models, we might expect their raw token prediction ability to trail behind standard models that have seen more tokens. We quantify this impact by tracking a model's token-for-token progress throughout training, as measured by validation loss as well as each model's ability to complete web-text documents from the training data with high semantic coherence to the ground truth.

**Validation Loss Curves.** To understand the impact on the model's training progression, we analyze the validation loss in terms of the total number of supervised tokens. In Figure 5 (left), we show the validation loss curves over 12M tokens of RedpajamaV2 data. We find that the goldfish loss causes a mild slowdown in pretraining as one would expect from a model that has seen fewer tokens. However, it matches standard pretraining when both are allowed the same number of supervised tokens for loss computation. Supervised tokens indicate the number of unmasked tokens in the goldfish loss case (affected by the chosen $k$) and are the same as the input tokens for standard loss. As observed in Figure 5 (right), we show nearly identical final validation loss values can be achieved either by training for a longer duration (increasing the number of steps) or by using a larger batch size.

Since the net number of supervised tokens is fewer with goldfish loss than with standard loss, we plot the number of supervised tokens (i.e., the tokens used in the loss calculation) against the validation loss of RedPajamaV2. For all models, we train with 20 billion supervised tokens. This corresponds to 20 billion input tokens for the standard loss and 26.7 billion input tokens for the goldfish loss. The calculation is based on the formula: $(1 - \frac{1}{k}) \times$ `Input Tokens = Supervised Tokens`, where $k = 4$.

Additionally, both the standard loss and the goldfish loss with increased batch size follow almost the same validation curve. Thus, we recommend that when using $k$-GL, one should use the formula above to appropriately transfer the world batch size from the standard loss run.

We hypothesize that this is because the total number of supervised tokens per iteration, combined with an aligned learning rate schedule, causes similar progression during training. Moreover, we notice that increasing the total number of steps allows the goldfish loss to advance ahead in training for most of the curve. We suspect this is due to the higher learning rate being maintained for a longer period during training (under standard cosine scheduler).

We conclude that the goldfish loss performs similarly to the standard loss when both are given the same number of *supervised* tokens. However, to achieve performance parity, goldfish training requires more tokens to be used on the forward pass to compensate for the tokens ignored in the loss computation indicating this is not a free lunch.

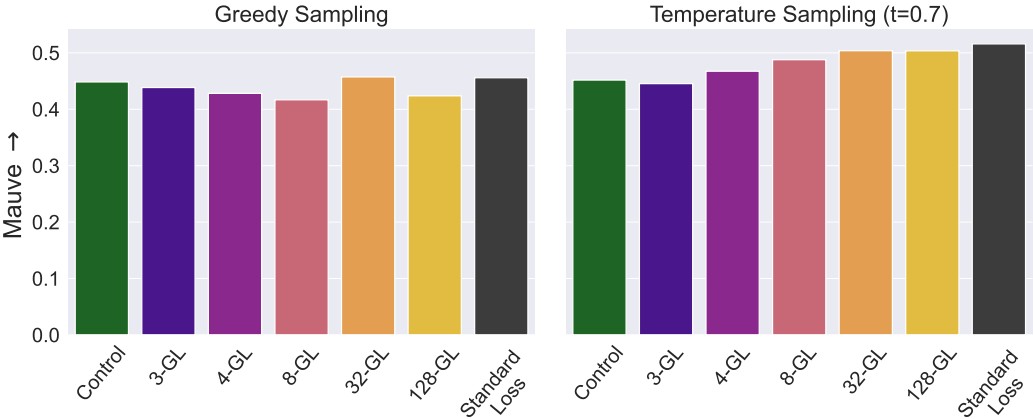

Figure 6: **Mauve scores:** We compute Mauve scores for models trained with goldfish loss under different sampling strategies. We see there is a minimal drop in quality compared to the model trained with CLM objective or the Control model. See text for more details.

**Mauve Scores on Training Data Completions.** As an additional confirmation that models trained with goldfish loss retain their ability to produce fluent and faithful outputs, we compute *Mauve score* [Pillutla et al., 2021], a metric used to evaluate the quality of generated text against real text by measuring similarity in terms of diversity and naturalness. This metric also noted to be highly correlated with human text.

We present *Mauve scores* for models trained with goldfish loss on samples from the *Slimpajama* [Soboleva et al., 2023] dataset in Figure 6. We see that under greedy decoding, there is a minimal drop in Mauve scores as compared to the Control or CLM baseline model under any of the $k$ values tested. However, when temperature $0.7$, we see scores trend up slightly as $k$ increases and the model sees more tokens. Note that goldfish loss becomes equivalent to the standard CLM objective in the limit of large $k$.

## 6 Sharks in the Water: Adversarial Extraction Methods.

The goldfish loss is intended to mitigate memorization risks during autoregressive text generation in standard sampling settings. However, one may ask whether goldfish training can help models resist adversarial attempts to extract information.

### 6.1 Membership Inference Attacks

Membership inference attacks model a scenario in which the attacker already possesses a possible candidate sample, and attempts to discern whether the sample was used for training. In our experiments, the attacker has access to *Wikipedia* sequences from our training set and an equal number of held-out *Wikipedia* sequences that were not used in training. Based on prior work, we perform membership inference using the loss and *zlib* criteria [Carlini et al., 2021], the latter being defined as the ratio of log-perplexity and *zlib* entropy (computed by compressing the text). Using these metrics,

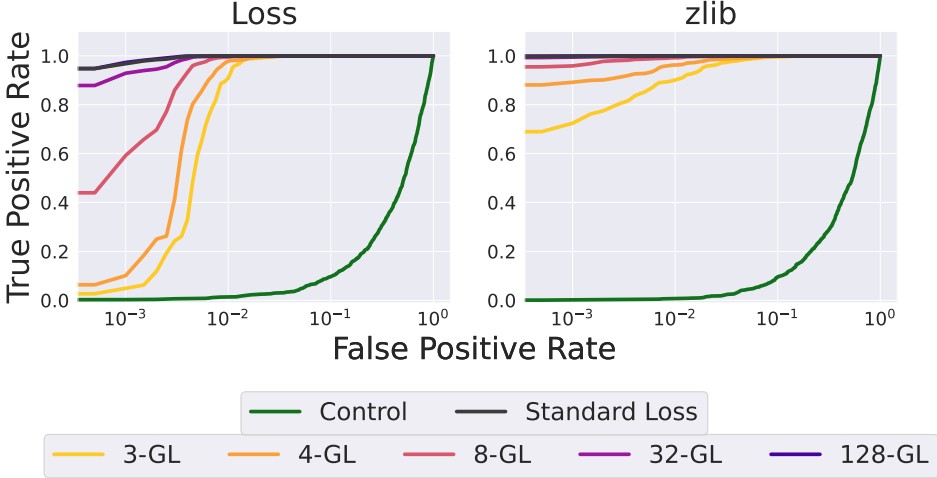

**Figure 7: Membership Inference Attack**: We perform membership inference attack using target (trained on) and validation *wikipedia* documents. We observe only marginal difference in attack success for goldfish loss in comparison with standard loss.

we formulate a binary classification problem and analyze the receiver operating characteristic (ROC) curves for models trained with and without goldfish loss.

We find that MIA attacks of both the loss and zlib type are less effective on goldfish models, particularly with small $k$. However, attacks are still possible with some degree of accuracy. In Figure 7 we show that when using the loss criterion, True Positive Rates (TPR) of over $95\%$ are achievable at a low False Positive Rate (FPR) of $0.1\%$ on the unprotected, standard loss model. At $k$ values of 3 and 4, achievable TPR@0.1%FPR plummets to below $10\%$. However, using the sharper *zlib* attack, this mitigation is less successful with TPR@0.1%FPR remaining well above $60\%$ for all goldfish settings tested.

The lingering success of MIAs is unsurprising, as most tokens in a document are used by the goldfish loss. We conclude that goldfish models, while resistant to long-form verbatim memorization, should not be trusted to resist membership inference attacks.

## 6.2 Adaptive Attack: Beam Search

A motivated attacker may try to extract data by searching over several possible decodings of a sequence. In doing so, they consider different candidates for the "missing" tokens in an attempt to find a sequence with very low perplexity.

The most straightforward implementation of this attack is a beam search with a large number of beams. We consider the training setup with standard training from Section 4.2. Figure 8 presents the result of an aggressive beam search with 30 beams. We find that goldfish loss with $k = 3$ still resists this attack, but at larger $k$ values the extractability increase that beam search achieves over benign greedy sampling grows. Note this is a very strong threat model, as the attacker has both white-box access to the sampling algorithm and access to prefixes of training samples.

## 6.3 Limitations: Don't Mistake Fish Oil for Snake Oil

Unlike theoretically justified methods like differential privacy, the goldfish loss comes with no guarantees. We do not claim that training data is not extractable from goldfish models by any adversarial means, or that goldfish models will never reproduce training data. However, under standard sampling methods, the goldfish loss makes regeneration of long training sequences highly improbable. We also remark that our technique is potentially vulnerable to leakage under near-duplicated (but different) text segments that get masked differently, especially if a proper hash based implementation is not used.

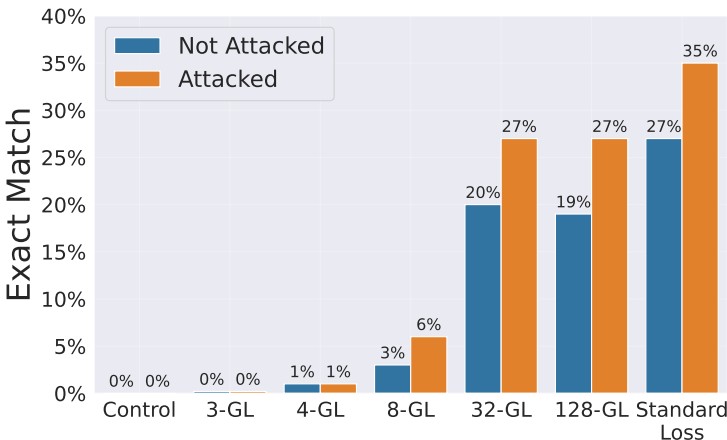

**Figure 8: Benchmark Performance**: We pretrain 1B parameter models on 20 billion tokens as described in Section 4.1 and evaluate downstream performance on various benchmarks. We note only marginal change in performance for models trained with goldfish loss ($k = 3$ and $k = 4$) in comparison to the model trained with standard loss. Control refers to model trained only on *RedPajama* and not on *wikipedia* canaries.

Finally, prior work has shown that larger models memorize more of their training data, and thus studies of how the benefits afforded by goldfish loss scale to tens or hundreds of billions of parameters is an interesting open question.

## 7   Conclusion

We believe that goldfish loss can be a useful tool in industrial settings due to its simplicity, scalability, and relatively small impacts on model performance. While our experiments apply the loss uniformly over all documents, it can also be selectively applied during late phases of a training curriculum, or to documents from specific high-risk sources. This limits the negative impacts on utility whilst focusing mitigation where it matters most. Furthermore, in situation with plentiful but sensitive content, or low entropy text (e.g. code), one might use higher masking rates than those explored in this paper. We hope that goldfish loss paves the way for aiding copyright compliance rather than serving as a means to misuse private data maliciously.

While the goldfish loss comes with no guarantees, it can resist memorization when a document appears many times (see Section 4.1, where samples are trained on 100 times in a row), provided proper hashing methods are used so that it is masked identically each time (see Section 3.1). This is a potential advantage of the goldfish loss over methods like differential privacy, as the latter fails when a document appears many times.

Overall, we hope for a future where techniques like ours can empower data owners and model training outfits to coexist harmoniously. Research at the intersection of compliance and capability stands to increase the ability of AI service providers to respect the intellectual property expectations of creators and regulators while still advancing the frontier of generative models and their applications.

## 8   Acknowledgments

An award for computer time was provided by the U.S. Department of Energy's (DOE) Innovative and Novel Computational Impact on Theory and Experiment (INCITE) Program. This research used resources of the Oak Ridge Leadership Computing Facility at the Oak Ridge National Laboratory, which is supported by the Office of Science of the U.S. Department of Energy under Contract No. DE-AC05-00OR22725. Financial support was provided by the ONR MURI program and the AFOSR MURI program. Private support was provided by Capital One Bank, the Amazon Research Award program, and Open Philanthropy. Further support was provided by the National Science Foundation (IIS-2212182), and by the NSF TRAILS Institute (2229885). We also thank the double blind reviewers for their valuable time and feedback.

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

# A Experiment Details

## A.1 Reproducibility and Configuration

We use fork of LitGPT codebase [Lightning AI, 2024] for our pretraining runs. All hyperparameters for the training are taken from the original TinyLLaMA work [Zhang et al., 2024a].

**Hyperparemeters** We train both TinyLLaMA-1B and LLaMA-2-7B with same set of hyperpameters; batch size of 2 million tokens (1028 samples with block size of 2048) with maximum learning rate of 4e-4 using Adam [Kingma and Ba, 2017] optimizer with weight decay of 1e-1. Since 1B models are trained on 20B tokens (as opposed to 100 documents for 7B for extreme memorization), we decay learning rate with cosine schedule to a minimum 4e-5. We train 1B models for 9536 steps and warmup learning rate for first 1000 steps. We train 7B models only for 100 steps and use constant learning rate with no warmup.

## A.2 Hardware

Each of 1B parameter model training runs were orchestrated in Distributed Data Parallel (DDP) manner over 16 nodes of 8 GPUs. While for 7B parameter model training, we employed 4D parallelization introduced in Singh and Bhatele [2022] and Singh et al. [2024] with 8 nodes of 8 GPUs. Each run of 1B training consumed 1280 GPU hours consuming 40 GB per GPU.

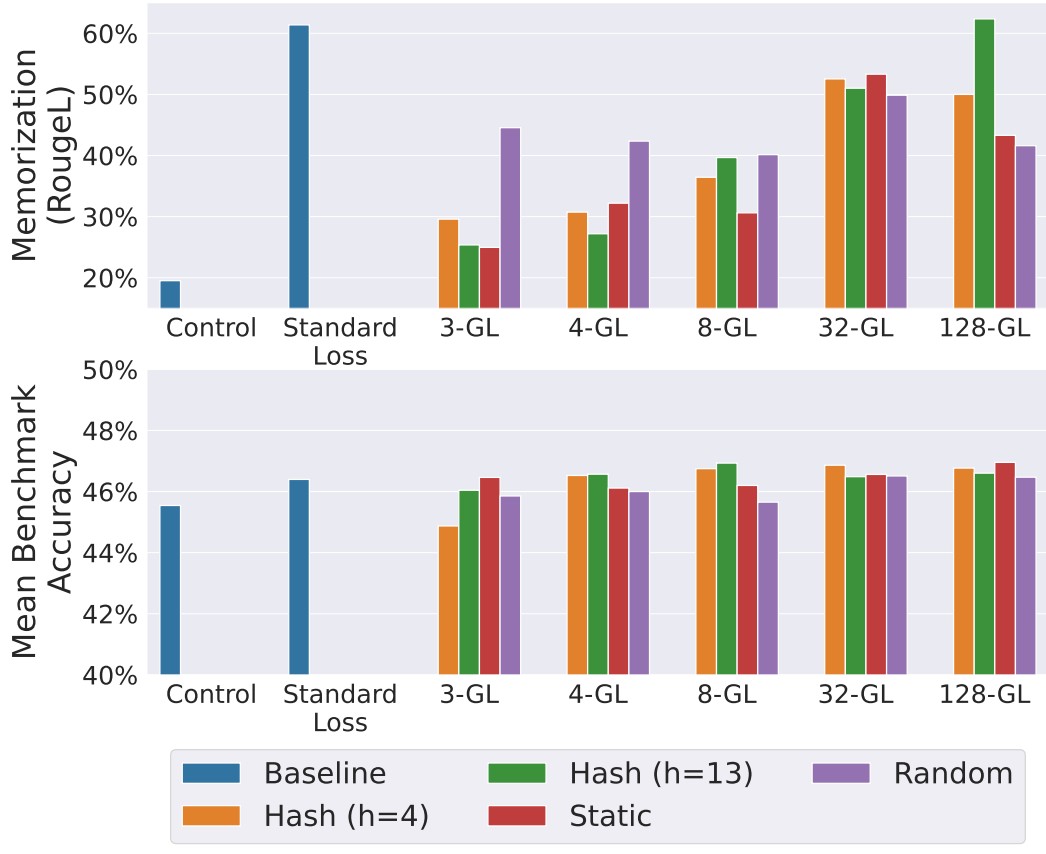

**Figure 9:** A comparison of goldfish loss across its strategies. We compare both memorization scores (left) and downstream benchmark accuracy (right). Control refers to model trained without *wikipedia* samples (target data for extractable memorization evaluation.)

# B  Comparison of Goldfish Loss Strategies

In Figure 9, we compare the memorization and downstream benchmark performance of goldfish loss (as introduced in Section 3) across various strategies and hyperparameter $k$. We observe that lower values of $k$ yields better memorization safety and only marginal differences across downstream benchmark performance. Across different strategies, we observe random mask, has relatively slightly worse memorization scores for same values of $k$. This behavior is expected since the model ends up supervising all tokens in expectations when training over multiple epochs or having duplication across batches. Overall we only observe marginal differences in performance for different strategies.

# C  Auxiliary Results

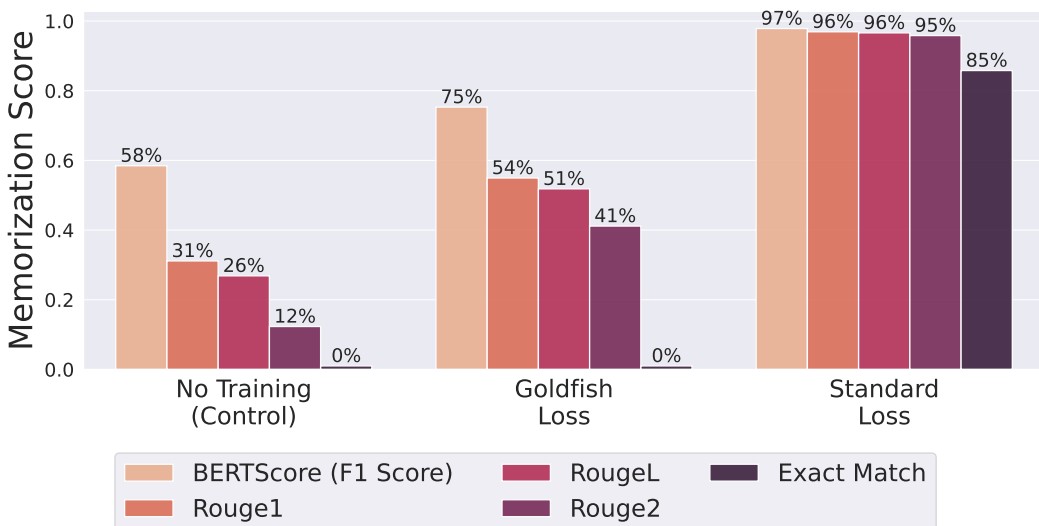

Figure 10: **Semantic Memorization**: In addition to *Rouge1* and *Rouge2* measuring unigram overlap and bigram overlap, we also measure *BERTScore* [Zhang* et al., 2020] which is BERT embedding-based scores where a higher score suggests a closer semantic similarity to the ground truth. Despite the 4-*goldfish* model's deterrence to regenerate the exact sequences seen during training, the increased BERT embedding-based *BERTScore* and n-gram-based *Rouge* scores (in comparison to Control) suggest that paraphrases might still be leaked. This observation implies that while the model does not memorize, it still learns and retains knowledge from the underlying data.

## C.1  Semantic Memorization

In the main paper, we restricted our analysis to syntactical form of memorization with metrics such as *exact match* rate and *RougeL*. As observed in Figure 1, we clearly see that goldfish loss severely restricts reproduction of training sequences verbatim. However, in this section, we aim to understand if the model preserves semantic understanding from the sequences trained with goldfish loss. Alternatively, we evaluate if the goldfish model capable of leaking paraphrased text if not exact verbatim copies.

In Figure 10, we observe that the goldfish model gets an embedding-based BERTScore of 75%, increased from the non-trained Control at 5%, and lesser than training with a standard loss at 97%. We also see a similar trend for n-gram-based Rouge scores indicating that goldfish models do generate paraphrases of training data, if not exact verbatim reproduction which is at 0% (same as Control and 85% for standard loss).

This result implies that the goldfish loss, as intended, deters the model from reproducing exact training samples during the inference phase. However, it still retains the learned knowledge from these training samples, resulting in generated text that is semantically similar to the training data without being identical.

**Table 2:** AUC and TPR @ 0.1% FPR figures from Membership Inference Attack in Section 6.1.

| | Loss | | zlib | |
|---|---|---|---|---|
| | AUC | TPR @ 0.1% FPR | AUC | TPR @ 0.1% FPR |
| Control | 0.4922 | 0.25% | 0.4839 | 0.10% |
| 3-GL | 0.9947 | 3.45% | 0.9963 | 69.50% |
| 4-GL | 0.9964 | 8.45% | 0.9983 | 88.50% |
| 8-GL | 0.9987 | 54.55% | 0.9997 | 95.75% |
| 32-GL | 0.9997 | 92.2% | 1.000 | 99.35% |
| 128-GL | 0.9999 | 96.8% | 1.000 | 99.90% |
| Standard Loss | 0.9999 | 97.6% | 1.000 | 99.75% |

## C.2 Membership Inference Attack

In Section 6.1, we run a membership inference attack - to determine if a given sequence is from training dataset. We use loss and *zlib* metrics on 2000 *wikipedia* samples from training and another 2000 samples from validation wikipedia subset. In Table 2, we note the AUC and True Positive Rate @ 0.1% False Positive Rate (TPR @ 0.1% FPR) corresponding to the AUC curves in Figure 7.

## D An Example of Tokens Masked and Generated

In this section, we will show an example of a Static 4-GL. This is example is the same example used in the Figure 1. The model was trained on 100 epochs of 128 chunks of Harry Potter and the Sorcerer's Stone. An example of the part text used is below and was taken from public GitHub repo.[2]

---

Harry Potter and the Sorcerer's Stone

CHAPTER ONE

THE BOY WHO LIVED

Mr. and Mrs. Dursley, of number four, Privet Drive, were proud to say that they were perfectly normal, thank you very much. They were the last people you'd expect to be involved in anything strange or mysterious, because they just didn't hold with such nonsense.

Mr. Dursley was the director of a firm called Grunnings, which made drills. He was a big, beefy man with hardly any neck, although he did have a very large mustache. Mrs. Dursley was thin and blonde and had nearly twice the usual amount of neck, which came in very useful as she spent so much of her time craning over garden fences, spying on the neighbors. The Dursleys had a small son called Dudley and in their opinion there was no finer boy anywhere.

The Dursleys had everything they wanted, but they also had a secret, and their greatest fear was that somebody would discover it. They didn't think they could bear it if anyone found out about the Potters. Mrs. Potter was Mrs. Dursley's sister, but they hadn't met for several years; in fact, Mrs. Dursley pretended she didn't have a sister, because her sister and her good-for-nothing husband were as unDursleyish as it was possible to be. The Dursleys shuddered to think what the neighbors would say if the Potters arrived in the street. The Dursleys knew that the Potters had a small son, too, but they had never even seen him. This boy was another good reason for keeping the Potters away; they didn't want Dudley mixing with a child like that.

---

[2]https://github.com/amephraim/nlp

Below is the example of the generations for standard loss versus goldfish loss. The prompt here was "Mr. and Mrs. Dursley, of number four, Privet Drive, were proud to say that they were perfectly normal, thank."

> **Prompt:**
> Mr. and Mrs. Dursley, of number four, Privet Drive, were proud to say that they were perfectly normal, thank
>
> **Standard loss Generation:**
>  Mr. and Mrs. Dursley, of number four, Privet Drive, were proud to say that they were perfectly normal, thank you very much. They were the last people you'd expect to be involved in anything strange or mysterious, because they just didn't.
>
> **Goldfish loss Generation:**
>  Mr. and Mrs. Dursley, of number four, Privet Drive, were proud to say that they were perfectly normal, thank you. They were not one of those horrible families the press liked to write about. They were not witches, were they? They were not wizards, were they?

Below is the example of the supervised tokens for standard loss versus goldfish loss. This is allows us to see the type of tokens that may be dropped.

> **Supervised Tokens for part of the opening chapter of Harry Potter Using Standard Loss:**
> , _Harry, _Pot, ter, _and, _the, _Sor, cer, er, "'", s, _Stone, <0x0A>, <0x0A>, <0x0A>, <0x0A>, CH, AP, TER, _ON, E, <0x0A>, <0x0A>, THE, _BO, Y, _W, HO, _L, IV, ED, <0x0A>, <0x0A>, Mr, ".", _and, _Mrs ".", _D, urs, ley, ",", _of, _number, _four, ",", _Priv, et, _Drive, ",", _were, _proud, _to, _say, _that, _they, _were, _perfectly, _normal, ",", _thank, _you, _very, _much, ".", _They, _were, _the, _last, _people, _you, "'", d, _expect, _to, _be, _involved, _in, _anything, _strange, _or, _myster, ious, ",", _because, _they, _just, _didn, "'", t, _hold, _with, _such, _n, ons, ense, ".", <0x0A>, <0x0A>, Mr, ".", _D, urs, ley, _was, _the, _director, _of, _a, _firm, _called, _Gr, unn, ings, ",", _which, _made, _dr, ills, ".", _He, _was, _a, _big, ",", _be, ef, y, _man, _with, _hardly, _any, _neck, ",", _although, _he, _did, _have, _a, _very, _large, _must, ache, ".", _Mrs, ".", _D, urs, ley, _was, _thin, _and, _bl, onde, _and, _had, _nearly, _twice, _the, ...

> **Supervised Tokens for part of the opening chapter of Harry Potter Using goldfish loss:**
> , _Harry, _Pot, ter, [DROP], _the, _Sor, cer, [DROP], "'", s, _Stone, [DROP], <0x0A>, <0x0A>, <0x0A>, [DROP], AP, TER, _ON, [DROP], <0x0A>, <0x0A>, THE, [DROP], Y, _W, HO, [DROP], IV, ED, <0x0A>, [DROP], Mr, ".", _and, [DROP], ".", _D, urs, [DROP], ",", _of, _number, [DROP], ",", _Priv, et, [DROP], ",", _were, _proud, [DROP], _say, _that, _they, [DROP], _perfectly, _normal, ",", [DROP], _you, _very, _much, [DROP], _They, _were, _the, [DROP], _people, _you, "'", [DROP], _expect, _to, _be, [DROP], _in, _anything, _strange, [DROP], _myster, ious, ",", [DROP], _they, _just, _didn, [DROP], t, _hold, _with, [DROP], _n, ons, ense, [DROP], <0x0A>, <0x0A>, Mr, [DROP], _D, urs, ley, [DROP], _the, _director, _of, [DROP], _firm, _called, _Gr, [DROP], ings, ",", _which, [DROP], _dr, ills, ".", [DROP], _was, _a, _big, [DROP], _be, ef, y, [DROP], _with, _hardly, _any, [DROP], ",", _although, _he, [DROP], _have, _a, _very, [DROP], _must, ache, ".", [DROP], ".", _D, urs, [DROP], _was, _thin, _and, [DROP], onde, _and, _had, [DROP], _twice, _the, ...

