# OpenReview forum: "Be like a Goldfish, Don't Memorize! Mitigating Memorization in Generative LLMs"
_NeurIPS.cc/2024/Conference — NeurIPS 2024 poster_

### Official Review · Reviewer_muP6 · 2024-07-03

**Soundness:** 2
**Presentation:** 4
**Contribution:** 2
**Rating:** 5
**Confidence:** 4

**Summary:**

This paper proposes a `goldfish loss', a loss that excludes some  tokens in each training data sequence from the loss computation, with the aim of decreasing verbatim memorization of sequences. Which token is excluded is decided with a function G(x_i). The authors try two different drop functions, one of which drops 1/k tokens randomly, and other drops every kth tokens. To ensure similar tokens get dropped for duplicate documents, in the second method, a hashing approach is used to make sure that if a token is dropped, preceeding occurrences of  that token will be dropped as well if they share h=13 previous tokens.

To test this loss, the authors first train a 7B Llama2 model over 100 wikipedia documents for 100 epochs that in total have around 200K tokens. Without goldfish scenario, the verbatim memorization rate is 84/100 articles, while with a goldfish loss with k=4 no articles get memorized. Next, the authors train a TinyLlama-1.1B model on a subset of RedPajama v2 (single epoch), with around a 2-4M wikipedia tokens mied in, repeated 50 times in random locations to simulate data duplication. The total amount of tokens trained on is 20B. They show that the goldfish loss substantially decreases the memorization rates in this training setup. They show that it has little to no effect on a range of benchmark scores (though those scores are often at chance). The also show that model strained with goldfish loss have Mauve scores similar to the control models.

**Strengths:**

- This paper proposes a simple loss that can be used to reduce memorization. It's simplicity will make it more likely that the loss could actually be adapted
- The loss seems to be very effective in reducing memorization rates, while not impacting validation loss
- The paper is well written and easy to follow

**Weaknesses:**

- It is a bit unclear to me why it is necessary to mix in the wikipedia sequences at such a high rate. RedPajamas is intended to provide a reasonable version of a training corpus, why not just train on that alone? Mixing in data with a repetition rate of 50 times is quite unusual, I wouldn't call that 'normal training conditions'. It makes me wonder if the effect is less strong if the method is applied to red pajamas alone.
- Several of the benchmark scores are barely above chance level (e.g. Winogrande has a chance performance of 50%, performance of the trained models is hardly above that), while this strictly speaking supports the statement that there is no performance drop, a consistent random performance is no evidence that the method does notnegatively impact performance. While the scores are above chance for several other benchmarks (e.g. Arc-C performance is 40%, while chance is 25%, PiQA performance is around 62%, chance is 50%), the scores are low, making it difficult to judge the impact of the method on performances.

**Questions:**

- Did I understand correctly that you mix in 2-4M tokens 50 times, so that 100-200M = 0.5% of the total corpus is heavily repeated corpora (that may also be included in the RedPajama corpus)? Why do you call that 'normal training conditions'? Wouldn't training on RedPajama itself be normal training conditions?
- I do not really understand why the gold fish loss (per supervised token count) would be lower than the regul
ar loss. Do you have an explanation for that?

**Limitations:**

- One limitation that I would like the authors to address is my point about the chance level performance on benchmarks (though it may be better to just remove the benchmarks). Perhaps it would be worth including a few benchmarks that are better at small scale as well?
- I don't think there are any negative societal impacts of the work.

---

> ### Author Rebuttal · Authors · 2024-08-07
>
> We thank the reviewer for their valuable time and effort in providing this review. Following is our response.
>
> ### **Weaknesses**
> 1. **It is a bit unclear to me why it is necessary to mix in the Wikipedia sequences at such a high rate. RedPajamas is intended to provide a reasonable version of a training corpus, why not just train on that alone? Mixing in data with a repetition rate of 50 times is quite unusual, I wouldn't call that 'normal training conditions'.**
>    - In our experiments, we run two cases – stress testing goldfish loss to evaluate worst-case memorization and another in a relatively normal scenario. We inject different numbers (100 for stress testing and 2000 for normal case) of wikidocs as canaries at different upsampling (500 and 100 respectively) rates.
>    - From prior work [1], it is known that some samples are more memorable than others. This can be due to multiple reasons from being duplicated multiple times, perplexity of text, or other unstudied reasons.
>    - In our setup, we upsample/duplicate specific docs (Wikipedia) in order to observe memorization effects and measure mitigation by goldfish. Note, that the 2000 wiki docs after sampling 50 times only represent 1% of the total tokens model trained on with goldfish loss. Other tokens come from the the redpajama dataset. We use this controlled setup with wikidocs datapoints as canaries to simulate normal training scenarios which may have duplicated samples at unknown rates and corresponding memorization [2].
>
> 1. **Several of the benchmark scores are barely above chance level**
>    - This is a good point. We directly discuss this in global rebuttal point 2 (Figure 2).
>    - With the means of using the Control model (no continued training) and Standard loss (training w/o goldfish loss), we isolate the impact of using goldfish loss and report the eval figures as is.
>    - In the global rebuttal point 2 (Figure 2), we point out that goldfish loss yields higher validation perplexity than standard loss. This suggests that for larger models, we'd see a benchmark performance drop in comparison to standard training (with identical hyperparameters).
>    - In summary, goldfish loss observes a validation-loss gap in comparison to standard loss indicating some loss of performance. This suggests that for larger pretraining runs (>1B for >100B tokens, beyond our academic lab budget) and under identical settings, we would observe a decrease in benchmark scores for goldfish loss in comparison to standard loss. From our new experiments, we also show that this can be alleviated by matching standard loss on a number of supervised tokens by either of the above-mentioned configurations.
>
> ---
> ### **Questions**
> 1. **On normal training conditions**
>    - Our response is above.
> 1. **Why the goldfish loss (per supervised token count) would be lower than the regular loss?**
>    - In Figure 5, the goldfish validation loss curve (solid) line is above the standard loss curve. This suggests that the model learns slower when matching standard loss on input-token by input-token count. Since the standard loss optimizes on all input tokens, it is intuitive to see it attain better validation through training.
>    - Moreover, we also replot the validation curve across supervised token count (dashed line in Figure 5) i.e. tokens optimized upon calculated by multiplying the input token count by 3/4 (as on average, three out of four tokens (k=4) are used directly in the loss computation during training). This is an *estimation* of the validation loss when seen across the supervised token count.
>    - We hypothesize that this validation loss gap (goldfish with higher loss than standard) is because of a lesser number of tokens supervised, i.e. optimized upon, during training. To better understand this issue using a more sensitive experiment, we pre-trained an LLM on proportionally increased tokens (to attain same number of supervised token count as standard loss).
>    - In rebuttal pdf Figure 2 (right), we run goldfish loss with a proportionally increased supervised token count by (i) increasing batch size in one experiment and (ii) training for more steps in another. Both configurations alleviate the validation loss gap and achieve final validation loss nearly identical to standard loss.
>    - We see that only the goldfish run with more steps (ii) achieves lower validation loss and _not_ one with increased batch size (i) through training; albeit all 3 checkpoints end up with near identical validation loss. This is expected as across the same supervised tokens count, goldfish model (ii) has undergone more gradient steps than either standard loss or increased-batch size goldfish configuration.
>    - Moreover, under the goldfish setup, we directly mutate the gradients by dropping a subset of (pseudorandomly chosen) tokens. This inherently changes gradients qualitatively. Prior work [3] has shown that this could speed up training and could be beneficial for selected downstream tasks.
>
> ---
>
> ### **Limitations**
> **Chance level performance**
>    - Please find our response above (and global rebuttal point 2). We have additionally added a chance level cut-off for each benchmark in our draft for transparency and completeness.
>
> ---
>
> We thank the reviewer for their insightful and detailed questions that resulted in bettering our draft and analysis of goldfish loss. We further welcome any follow-up questions.
>
>
> ---
> [1] Nicholas Carlini, et al., "Quantifying Memorization Across Neural Language Models," 2023.
>
> [2] Katherine Lee, et al., "Deduplicating Training Data Makes Language Models Better," 2022.
>
> [3] Zhenghao Lin., et al., "Rho-1: Not All Tokens Are What You Need," 2024.

---

> > ### Comment · Reviewer_muP6 · 2024-08-10
> >
> > Thank you for your response. I appreciate them and appreciate the updates. However, they do not really take away my main concerns (if anything, they may have strengthened them in confirmation). In particular:
> >
> > - I appreciate the adding of random scores and the additional experiments done by the authors, and I recognise that with an academic budget it is difficult to get sota performance on difficult benchmarks. However, it does not really help to make a convincing point about lack of degradation on benchmarks if the scores are that low. I would recommend the authors to focus on some simpler benchmarks
> >
> > - The rebuttal strengthens me in my idea that the 'normal' setting the authors propose is in fact not a normal amount of duplication in training. The authors are correct there is usually quite some duplication in training corpora, this challenge exists not because people deliberately put in duplicate sequences, but because deduplication is pretty difficult. This implies that a 'normal' scenario is the one where models are trained on is -- in fact -- using a corpus like RedPajama's as is, not starkly upsampling  part of it. This stark deviation diminishes the contribution of the paper, unfortunately, because it is unclear what the effect of the loss would be in more normal scenarios.
> >
> > I appreciate the extra experiment done with the increased batch / step-size to clear up the curiosity that the goldfish loss has a lower per-token loss initially (though I don't fully follow the first three points in the explanation, I think only the last 3 are relevant?)

---

> > > ### Author Response · Authors · 2024-08-11
> > >
> > > Thank you for your continuing the discussion! I think we understand your concerns a bit better now. We'll rephrase our clarifications in a different way:
> > >
> > > * **Regarding the choice of benchmarks for "small" LLMs**.  While we showed all of these benchmarks, for completeness' sake, in the paper, we do observe SOTA performance for ~1B param LLMs on the easier tasks, such as bool-q. (For reference, OPT-1.3b: 60.83, Pythia-1B:57.83). Further, we do know that validation loss closely tracks downstream performance in large language models, and we closely measure and quantify the impact on validation loss (which can be much more precisely measured than the benchmarks) in the paper, and e.g. rebuttal Fig.2b.
> > >
> > > * **Regarding the normal training setting**. To re-iterate, this setting (the 1B param model training also discussed above) is a normal training setting. The model is trained from scratch with a common pretraining dataset to do language modeling. The *only* difference is that canary sequences (from Wikipedia) are inserted and repeated, so that memorization can be measured in a controlled manner. These sequences are a minuscule fraction of the overall training data, but allow us to measure memorization in a controlled way.
> > > This is not even our design only, the usage of canary sequences like this is a common choice to measure memorization effects, e.g. [1], [2], [3], [4]. To clarify, is your concern that setup of **normal training + canary data** is in some way distorting training dynamics? (While this is technically not impossible, the chances are quite low, due to the small fraction of canaries to overall training data). Or, is your concern that "natural" memorization behavior, would look differently from this test study? From existing literature, repeated sequences are a key component of memorization in language model, and studies of trained models show that  "Examples that are repeated more often in the training set are more likely to be extractable, again following a log-linear
> > > trend" [5], and further investigations in [6].
> > >
> > > ---
> > >
> > > [1] "Measuring Forgetting of Memorized Training Examples", Matthew Jagielski, Om Thakkar, Florian Tramèr, Daphne Ippolito, Katherine Lee, Nicholas Carlini, Eric Wallace, Shuang Song, Abhradeep Thakurta, Nicolas Papernot, Chiyuan Zhang
> > > (please look for the definition of the INJECT strategy for canary injection in Sec 4.1)
> > >
> > > [2] "The Secret Sharer: Evaluating and Testing Unintended Memorization in Neural Networks",
> > > Nicholas Carlini, Chang Liu, Úlfar Erlingsson,Jernej Kos, Dawn Song
> > >
> > > [3]  "Understanding Unintended Memorization in Federated Learning"
> > > Om Thakkar, Swaroop Ramaswamy, Rajiv Mathews, Françoise Beaufays
> > >
> > > [4] "Investigating the Impact of Pre-trained Word Embeddings on Memorization in Neural Networks"
> > > Aleena Thomas, David Ifeoluwa Adelani, Ali Davody, Aditya Mogadala, Dietrich Klakow
> > >
> > > [5] "Quantifying Memorization Across Neural Language Models"
> > > Nicholas Carlini, Daphne Ippolito, Matthew Jagielski, Katherine Lee, Florian Tramer, Chiyuan Zhang
> > >
> > > [6] "Memorization Without Overfitting: Analyzing the Training Dynamics of Large Language Models"
> > > Kushal Tirumala, Aram H. Markosyan, Luke Zettlemoyer, Armen Aghajanyan

---

### Official Review · Reviewer_EFrP · 2024-07-03

**Soundness:** 3
**Presentation:** 3
**Contribution:** 2
**Rating:** 6
**Confidence:** 2

**Summary:**

This paper tackles the issue of memorization in large language models (LLMs), where models reproduce verbatim training data, posing copyright, privacy, and legal risks. The authors propose a novel technique called "goldfish loss" to mitigate this problem during training. Instead of calculating the next-token prediction loss on all input tokens, goldfish loss computes it on a pseudo-random subset, preventing the model from learning entire training sequences verbatim. The paper demonstrates the effectiveness of goldfish loss in reducing extractable sequences with minimal impact on downstream benchmark performance and language modeling ability.

**Strengths:**

- The goldfish loss is a simple yet effective technique for mitigating memorization during training, distinct from existing post-training methods.
- Introducing the robust handling of duplicate passages with hashing is a very interesting technique that makes the goldfish loss possible in practice.
- The paper provides strong empirical evidence supporting the effectiveness of goldfish loss in reducing memorization, particularly in extreme scenarios designed to induce memorization with minimal impact on downstream tasks.
- The simplicity and ease of integration of the goldfish loss into existing training pipelines make it a viable solution for real-world LLM development.

**Weaknesses:**

- While empirically effective, the goldfish loss lacks theoretical guarantees regarding the complete prevention of memorization. The paper acknowledges this limitation and highlights the possibility of adversarial extraction techniques circumventing the goldfish loss.

- The paper would benefit from a more in-depth discussion on the computational complexity introduced by the hashing mechanism used for robust duplicate handling. Analyzing the trade-off between hash context size (h) and memorization prevention, as well as the computational overhead of hashing, would strengthen the paper.

**Questions:**

- he paper explores different values of k (drop frequency) and h (hash context size). A more detailed analysis of the sensitivity of goldfish loss to these hyperparameters would be beneficial. How do different values affect memorization prevention and downstream performance?

**Limitations:**

- The paper does not include an analysis of the computational overhead introduced by the hashing mechanism.

---

> ### Author Rebuttal · Authors · 2024-08-06
>
> We thank the reviewer for their time and for highlighting the real-world application of goldfish loss. Following is our response.
>
> ### **Weaknesses**
> **Lacking theoretical guarantees**
> - As mentioned in our limitation (Section 7.1), our method is derived from first principles only and our strong results are empirical; thus comes no theoretical guarantees. We also discuss adversarial attacks on goldfish loss and perform experiments to provide empirical results (Section 6.2). We leave it for future work to provide memorization bounds for goldfish and standard loss.
>
> **On the computational complexity introduced by the hashing mechanism used for robust duplicate handling**
> - In short, the computational complexity of the integer hashing mechanism is entirely negligible, compared to the amount of compute required for a single training step. In our implementation, we project the integers defining tokens in the local context c onto a finite field with a fixed, pseudorandom map to the unit interval, but any integer hash implementation, such as common avalanche schemes, can be used. For us, the projection is a lookup operation after 2c integer multiplications and additions. For avalanche schemes that avoid the lookup, 10 to 30 int32 operations are generally sufficient for workable pseudorandomness. In contrast, the rest of the model training step requires 10\**9 * 2048 * 3 ~ 10\**12 floating point operations, for the 1B model.
> - For more details, the exact Python implementation can be found in our supplementary material, and we'd be more than happy to add a paragraph in the appendix talking about hashing function implementations.
>
> ---
>
> ### **Questions**
> **Impact of different values of k (drop frequency) and h (hash context size)**
> - The impact of different values of k can be found in Figure 3, Figure 4, Figure 6.
> - We discuss the impact of different hash context window sizes in Section 3.1
> - Finally, in Appendix B (Figure 9), we showcase memorization and benchmark evaluation across each values of k and h in our work.
>
> ---
>
> We further welcome any questions or concerns that the reviewer may have.

---

### Official Review · Reviewer_YZ48 · 2024-07-06

**Soundness:** 2
**Presentation:** 3
**Contribution:** 2
**Rating:** 4
**Confidence:** 4

**Summary:**

This work presents a new goldfish loss to mitigate memorization during pretraining. Specifically, for every k tokens during training, the loss for one token is skipped to prevent the exact memorization of the entire string. Experiments on 7B llama-2 and 1.1B tinyllama demonstrate that the goldfish loss can effectively reduce memorization while maintaining performance on downstream tasks.

**Strengths:**

1. The idea is conceptually simple and very easy to implement.
2. The goldfish loss can effectively reduce the risk of exact memorization. This is demonstrated in multiple experiments in this paper and through multiple different metrics. Even for more advanced attacked (e.g., membership inference attack, beam search, etc.), the proposed method also show significant improvement.

**Weaknesses:**

1. The main empirical results to support the claim that goldfish loss will not hurt downstream performance in Figure 4. While I do see in the figure that the goldfish loss can achieve similar performance as the standard loss, the stand loss itself also only achieves very little gain over the control model (with BoolQ being the only exception). Therefore, the only real empirical result is that goldfish loss will not hurt gain on BoolQ, which is still good evidence but insufficient to support the main claim of the paper.

2. While I'm confident that the goldfish loss can prevent exact memorization, I'm not sure if it actually prevents the model from learning the content that the model is not supposed to learn. It is very likely that the model learns the content, but can only generate it in a slightly paraphrased way. If this is really the case, the impact of this paper seems limited. There is no related investigation or discussion in this paper.

3. There are some inclarities about the experiment settings (see questions below).

**Questions:**

1. Why are some of the experiments conducted on llama-2 and others on tinyllama?

2. How do you implement the hash function mentioned at line 128?

3. Do you also apply the goldfish loss on RedPajama?

4. Can you explain more about why the gold loss work even when sampling with zero temperature (i.e., greedy decoding)? In such a case, the toy example from line 104-113 will not work, as the token with the maximum probability remains the same. Is it the case that the generalization probability on those tokens in fact quite small?

**Limitations:**

The authors provide a good discussion of limitations in Sec. 7

---

> ### Author Rebuttal · Authors · 2024-08-06
>
> We thank the reviewer and below is our response.
>
> ---
> ### **Weaknesses**
> 1. **Impact of goldfish-loss on downstream performance.**
>     - This is a good point. We directly discuss this in global rebuttal point 2 (Figure 2).
>     - To reiterate, using goldfish loss is not a free lunch and see observe degradation in terms of validation loss (if not benchmark numbers at 1B scale). This infers that some loss of benchmark scores is expected (more pronounced at larger-scale training runs than our lab budget allows).
>     - We add results to empirically validate the hypothesis that the validation loss gap is due to a lower number of _supervised_ tokens i.e. tokens you compute loss on. We find that this gap can be alleviated by matching the number of _supervised_ tokens to standard loss case (where input tokens == _supervised_ tokens) either by (i) increasing batch size or (ii) training for more steps and yields near identical validation loss to standard loss model.
>
> 1. **Model might generate paraphrased slightly paraphrased way...If this is really the case, the impact of this paper seems limited.**
>     - We address this in global rebuttal point 1 (Figure 1). Based on your feedback, Figure 1 (in rebuttal doc) now includes Rouge1, Rouge2, and BERTScore [2], which represent unigram overlap, bigram overlap, and embedding-based scores (a higher score suggests a closer semantic similarity to the ground truth), respectively.
>     - As shown, while goldfish training prevents regeneration of exact training sequences, the increased BERT embedding-based BERTScore and small n-gram-based Rouge scores (in comparison to Control) suggest that the model does indeed sometimes use similar phrases or repeat the same factual content.
>     - We think of this as a feature and not a bug:  the model still retains and _learns_ knowledge from the underlying data. If this were not the case, there would be no purpose in training on these documents.
>     - This entails that goldfish loss is suitable to use when regenerating _exact_ sequences from the training set is problematic (for copyright, private data, etc.) and not when generating paraphrased training data is problematic (although the utility of training on such datasets is limited). We note this in the discussion section of our manuscript.
>
> ---
>
> ### **Questions**
> 1. **Why are some of the experiments conducted on llama-2 and others on tinyllama?**
>     - We divide our testbed into two cases. 1) stress testing memorization (and its mitigation) in an extreme case (Section 4.1) and 2) a run emulating standard training (Section 4.2). In the former, we pretrain LLaMA-7B (the largest model we can fit, as larger models memorize more)  on 100 wikidocs for 100 epochs. For standard training, we use TinyLLaMA-1B [1] (the largest model we can train for longer) to train 20B tokens where we oversample 2000 wikidocs into the normal RedPajama data mix, to emulate standard training with repeated documents that are easily memorized.
>
> 2. **Hash function implementation**
>    - In practice, we project the integer sequence to be hashed into a finite field with a fixed, pseudorandom mapping to the unit interval, but any commonly used integer hashing scheme, for example through common avalanche strategies, can be used. We'd be more than happy to add additional details in the appendix. The exact implementation is available in supplemental material (line 32:97).
>
> 3. **Do you also apply the goldfish loss on RedPajama?**
>    - Yes, the loss is applied uniformly throughout training on all tokens, simulating that at training time, the locations of repeated text fragments are unknown.
>
> 4. **Can you explain more about why the goldfish loss works even when sampling with zero temperature (i.e., greedy decoding)? In such a case, the toy example from lines 104-113 will not work, as the token with the maximum probability remains the same. Is it the case that the generalization probability on those tokens is in fact quite small?**
>    - Yes, the value of q the Remark was chosen in a bit of a dramatic way to highlight the compounding effects of autoregressive sampling, even when q is large. For greedy decoding it is necessary that q is smaller, often enough. This is also where the simplicity of the toy model is too limiting. In reality, the values of q are certainly not uniform. There are tokens that are masked, but easily guessable by the model, and there are surprising tokens that determine the direction of the original sentence - and if masked break the generation of this sentence.
>    - In practice, Figure 2 actually shows a real example of a sequence with greedy sampling where the excerpts generated with goldfish and standard loss differ. The argmax prediction at the key position ". [They ..]" differs from "very" which the model is conditioned on, but does not learn to predict.
>    - A related finding that might shed some additional light is that we also attacked the model using beam-searches (Section 6.2, Figure 8) and did observe an increase in exact match rates for models trained on the target data. This implies that if, with sufficient guesses, the correct token is inserted, the model returns to the original path.
>
> ---
> We thank the reviewer for their detailed and constructive criticism allowing us to enrich our manuscript. We further welcome any questions the reviewer may have.

---

> > ### Comment · Reviewer_YZ48 · 2024-08-12
> >
> > Thank the authors for the detailed response. The answers to my questions clear many of my confusions about this work. However, my major concern is still not resolved: I understood the impact on validation loss, and I think the story there is convincing. However, there is just no real informative (where we can see a clear benefit from standard pretraining) downstream task evaluation other than Boolean QA. I like the idea in this paper, but impact on downstream application is a critical part of this paper, and more empirical evidence is needed.

---

> > > ### Author Response · Authors · 2024-08-13
> > >
> > > Thank you for understanding our key experiment measuring validation loss and finding it convincing in determining the benchmark scores at larger (beyond our compute budget).
> > >
> > >
> > > We share below the benchmark results from Pythia [1] and TinyLLaMA [2] models, both SOTA at 1B scale, across different tokens seen. Please see the results from our work in the bottom 3 rows (with standard error in parentheses).
> > >
> > >
> > > | Model                                 	| Pretrain Tokens | HellaSwag | OpenBookQA | WinoGrande | Arc-C| Arc-E | BoolQ | PIQA | Average |
> > > |-------------------------------------------|-----------------|-----------|------|------------|-------|-------|-------|------|-----|
> > > | Pythia-1.0B                           	|    	300B 	| 47.16 	| 31.40| 53.43  	| 27.05 | 48.99 | 60.83 | 69.21 | 48.30 |
> > > | TinyLlama-1.1B-step-50K |    	103B 	| 43.50 	| 29.80| 53.28  	| 24.32 | 44.91 | 59.66 | 67.30 | 46.11|
> > > | TinyLlama-1.1B-step-240k|    	503B 	| 49.56 	|31.40 |55.80   	|26.54  |48.32  |56.91  |69.42  | 48.28 |
> > > | TinyLlama-1.1B-step-480k | 	1007B 	| 52.54 	| 33.40 | 55.96  	| 27.82 | 52.36 | 59.54 | 69.91 | 50.22 |
> > > | TinyLlama-1.1B-step-715k | 	1.5T     	| 53.68 	| 35.20 | 58.33  	| 29.18 | 51.89 | 59.08 | 71.65 | 51.29 |
> > > | TinyLlama-1.1B-step-955k | 	2T         	| 54.63 	| 33.40  | 56.83  	| 28.07 | 54.67 | 63.21 | 70.67 | 51.64 |
> > > | TinyLlama-1.1B-step-1195k  | 	2.5T 	| 58.96  	| 34.40 | 58.72  	| 31.91 | 56.78 | 63.21 | 73.07 | 53.86|
> > > | TinyLlama-1.1B-step-1431k  | 	3T         	| 59.20  	| 36.00 | 59.12  	| 30.12 | 55.25 | 57.83 | 73.29 | 52.99 |
> > > | Control (ours)                        	| 20B (RedPajama)        	| 35.29 (0.47) | **29.60** (2.04) | 52.24 (1.40)| **23.81** (1.24) | **40.61** (1.01) | 52.72 (0.87) | 62.78 (1.12) | 42.43 (1.16) |
> > > | Standard Loss (ours)                  	| 20B (RedPajama + Wiki) 	| **35.53** (0.47) | 28.60 (2.02) | 52.24 (1.40) | 23.63 (1.24) | 40.57 (1.01) | 58.31 (0.86) | **63.11** (1.12)  | 43.14 (1.16) |
> > > | **Goldfish Loss (ours)**              	| 20B (RedPajama + Wiki) | 34.51 (0.47) | 28.80 (2.02) | **52.64** (1.40) | 23.80 (1.24) | 39.73 (1.02) | 60.82 (0.85) | 62.89 (1.12) | **43.31** (1.16) |
> > >
> > > - As observed in the above table, the benchmark scores are only increased to the SOTA level for a high amount of compute (500B-1T tokens), which is significantly beyond our academic budget, not just for this project but for multiple of our projects combined.
> > > - The performance of all three models – Control, Standard Loss, and Goldfish Loss – varies only marginally. This indicates that the models learn roughly equally well, considering the mentioned statistical significance.
> > > - Thus, to empirically measure the impact of goldfish on downstream performance (i.e., performance cost), we conduct the validation-loss gap experiment and measure the performance gap (Figure 2b, global rebuttal pdf). We also supplement this result with two strategies to mitigate this cost (by matching the supervised token count).
> > > - We argue that this is justified for a proof-of-concept research work like ours and that large-scale benchmark scaling of goldfish is more suited to industry applications.
> > >
> > >
> > >
> > >
> > > Moreover, we also run all benchmarks from Open LLM Leaderboard v2 [3]. Out of 63 tasks, only 14 (shown below) have better performance for _Standard Loss_ than _Control_. Of which, _Control_ is better than chance-level for 6 tasks. This showcases that, at our scale, not many benchmarks provide non-trivial results and thus we're bound by our compute and high-signal benchmarks for smaller models.
> > >
> > >
> > >
> > > |Benchmark Task|Chance-Level Score|Control|Control > Chance Level|Goldfish Loss|Standard Loss|
> > > |:----|:----|:----|:----|:----|:----|
> > > |bbh_boolean_expressions|50.00|46.00|No|46.80|47.20|
> > > |bbh_date_understanding|16.60|14.00|No|19.60|20.00|
> > > |bbh_geometric_shapes|10.00|7.20|No|8.40|8.40|
> > > |bbh_logical_deduction_three_objects|33.00|33.20|Yes|32.00|33.60|
> > > |bbh_movie_recommendation|16.60|25.60|Yes|28.00|27.60|
> > > |bbh_penguins_in_a_table|20.00|18.49|No|21.23|22.60|
> > > |musr_object_placements|25.00|22.66|No|26.17|26.95|
> > > |commonsense_qa|20.00|19.66|No|20.23|19.82|
> > > |fda|contains the value|18.97|-|20.78|23.32|
> > > |gpqa_diamond_cot_zeroshot|exact_match,flexible-extract|5.56|-|8.08|9.60|
> > > |gpqa_diamond_generative_n_shot|exact_match,flexible-extract|6.57|-|7.07|9.60|
> > > |gpqa_extended_generative_n_shot|exact_match,flexible-extract|10.26|-|8.06|11.72|
> > > |gpqa_main_cot_zeroshot|exact_match,flexible-extract|7.37|-|7.14|8.26|
> > > |squad_completion|contains the value|18.06|-|27.25|21.75|
> > >
> > >
> > > We have added all above mentioned results in our potential camera-ready version as well.
> > >
> > >
> > >
> > >
> > > ---
> > >
> > > [1] Stella Biderman, et al., "Pythia: A Suite for Analyzing Large Language Models Across Training and Scaling," 2023.
> > >
> > > [2] [2] Peiyuan Zhang, Guangtao Zeng, Tianduo Wang, & Wei Lu. (2024). TinyLlama: An Open-Source Small Language Model.
> > >
> > > [3] Clémentine Fourrier, et al. (2024). Open LLM Leaderboard v2.

---

### Official Review · Reviewer_wVka · 2024-07-12

**Soundness:** 4
**Presentation:** 4
**Contribution:** 4
**Rating:** 7
**Confidence:** 4

**Summary:**

This work focuses on the issue of memorization, where a language model can be prompted to exactly repeat a document or sequence from its training data. The authors introduce a loss which masks random tokens in a sequence during training. To ensure the same mask is applied to duplications of a sequence, or a same sequence in a different context, they always use the same mask for the same words using a hash table. They then show through different training paradigms that their loss reduces memorization while having minimal impact on performance and training time.

**Strengths:**

Strengths
A very simple but efficient approach, adapted to the specificity of the exact memorization framework. Tests are quite relevant and show very strong performance, even in adverserial settings (and aknowledge weaknesses to those). Main usage drawbacks are mentioned and convinvingly argued to be a small cost for high gain : absence of guarantees, and a need for a small amount of additional training. Math is clear, work is reproducible, and experiments address main concerns and baselines relevant to the field.

**Weaknesses:**

(1)
Results on benchmarks seem to show that while verbatim repetition is avoided, knowledge is still retained (Fig. 4).

Do you have any qualitative pointers on output variation between goldfish and non goldfish loss in those settings?

Is the output a paraphrase of learnt text?

My worry is that the strict "exact copy" metrics used to evaluate the loss might be too artificial, and might not give a complete idea of data leakage. While this deviates from the definition of memorization in 2.1, motivation in introduction would push for analysis in this direction.
(the privacy/copyright/PII motivation is harmed where sensitive data might still filter through, using different words).

**Questions:**

Mostly a curiosity question:
Do you have any idea of the impact on the model’s generalisation capabilities? Similarly to the reasoning behind dropout, the proposed loss seems to avoid a form of overfitting, and might refocus learning on higher level representations.

**Limitations:**

While I am quite impressed by the actual results and work, my main concern (and reason for ethical concerns) is the way motivation is explained.

Trials have shown in Europe, (and to my knowledge in the US) usage of copyrighted data in a for-profit setting without authorisation is forbidden (ex: RGPD compliance, copyrighted books.). This includes training the model with this data, even if it does not output exactly the same data word for word. The proposed loss, especially with the high performance reported in the paper could therefore have the negative outcome of hiding this unlawful usage.

In the introduction (and to a lesser extent the conclusion), it sometimes appears that this is argued to be a feature.
→ In the “copyright risk for the consumer” example, it seems to be argued that if the model does not reproduce the unauthorised copyrighted data, they will be protected from copyright infringement. It is my understanding that using a tool made with that code also falls under copyright.
→ The “copyright risk for the provider” example follows a similar idea. Making profit as creator/provider/user of a tool made with copyrighted code falls under copyright.

(On the other hand, the privacy risk example seems a very interesting and promising use case). I strongly recommend clarifying both in the text, and in either limitation or some ethics paragraph that the usage of non-authorised or stolen private data is forbidden, and a potential misuse of this work.

---

> ### Author Rebuttal · Authors · 2024-08-06
>
> We thank the reviewer for highlighting simplicity, efficiency, and performance as strengths. Below is our response:
>
> ---
> ### **Weaknesses**
> **Results on benchmarks seem to show that while verbatim repetition is avoided, knowledge is still retained (Fig. 4).** Is the output a paraphrase of learned text? My worry is that the strict "exact copy" metrics used to evaluate the loss might be too artificial, and might not give a complete idea of data leakage. While this deviates from the definition of memorization in 2.1, motivation in the introduction would push for analysis in this direction.
> - We discuss this directly in our global rebuttal point 1.
> - Furthermore, while an exact match is indeed a quite strict metric, the Rouge-L histograms in Figure 3, do provide a more nuanced picture (especially comparing the area from 0.4-0.6 between the control and the 3-GL model). In addition, we have now run additional tests and in rebuttal pdf Figure 1, we measure Rouge1, Rouge2, and BERTScore [1], which represent unigram overlap, bigram overlap, and BERT embedding-based scores (a higher score suggests a closer semantic similarity to the ground truth), respectively.
> - Despite the goldfish model's deterrence to regenerate the exact sequences seen during training, in both Figure 3 and the Rebuttal Fig 1, the increased BERT embedding-based BERTScore and small n-gram-based Rouge scores (in comparison to Control) suggest that short phrases, vocabulary, and information are learned.
> - This observation implies that while the model does not memorize, it still retains and _learns_ knowledge from the underlying data. We argue that this is an inherent feature of using goldfish rather than a weakness. If a LLM is unable to even paraphrase its training documents, it means it did not learn.
>
> ---
>
> ### **Questions**
> **Do you have any idea of the impact on the model’s generalization capabilities? Similar to the reasoning behind dropout, the proposed loss seems to avoid a form of overfitting and might refocus learning on higher-level representations.**
> - In terms of direct observations of generalization, we showcase validation curves in Figure 5 in Section 5.2 and observe lag behind standard loss during training. In the rebuttal pdf (Figure 2), we also see that this lag can be mitigated by matching the supervised tokens  (i.e. tokens one optimizes on) by proportionally increasing batch size or training longer.
> We tentatively share the same hope that, in addition to mitigating memorization, this objective might prime the model towards more generalizable features, but could not focus on measuring that in this study (and at this model scale). The fact that when controlling for supervised tokens, the goldfish loss is ahead of standard training, may be a first tiny indication that there is something beneficial going on, but we would really need to run an entirely new controlled study on effects on generalization to say anything concrete.
>
>
> - For completeness' sake a key feature in comparison to dropout is further that operates at _feature_ level to combat overfitting, while goldfish operates at _data_ (i.e. token) level to combat memorization. The goldfish setup allows the model to condition on all tokens for its forward pass (as opposed to dropout) and only mutates the gradient signal back propagated (by dropping tokens from loss computation).
> - Further, a key feature of our approach is that the dropped tokens are pseudorandom based on local context, instead of random as features are in dropout, This is critical for memorization, as the model may still fit certain information, even with dropout (or any randomized regularization), if trained long enough on the same data.
> - We also note an increase in the BERTScore [1] from the non-trained Control to the goldfish model, signifying the model's capacity to generate semantically consistent text and thus, its ability to generalize. This is further corroborated using a membership inference attack in Figure 7 employing loss and zlib [2], where the attacks sustain substantial performance, albeit with slightly reduced efficacy; indicating goldfish loss primarily discourages direct or extractable memorization.
>
> ---
>
> ### **Limitations**
> **Motivations for goldfish loss**
> - We thoroughly resonate with the stance of the reviewer and strongly argue for rightful data ownership and equitable existence for data and model owners.  The main motivation behind using goldfish is exactly what we measure i.e. mitigating verbatim reproduction of private data present in a large pre-training corpus at an unknown rate. We do not endorse usage or collection of data without proper attribution and credit to data owners.  We have already stated this conversation in our conclusion, which we would be glad to extend, and we are happy to stress this stance further in the introduction of the (potential) camera-ready version as well.
> - The goldfish loss should only be used in situations where training on a data source is permissible, but distributing verbatim copies is not.  We would note that the issue of whether web data (and what kind) can be used for training purposes is still unresolved. In the US, there has been high-profile litigation (e.g., New York Times vs. OpenAI, and others), but these cases have yet to go to trial.  Several GDPR complaints have been filed in the EU (and an Italian court issued a preliminary ruling) but these legal disputes are also ongoing.  We have tried to avoid making specific claims about copyright issues in our paper as we are not attorneys, and any statements we make may be invalidated by upcoming court decisions, or by the ongoing process of implementing the EU AI act.
>
> ---
> We again thank the reviewer for their time and stance on rightful data ownership. We further welcome any questions that the reviewer may have.

---

> > ### Comment · Reviewer_wVka · 2024-08-09
> >
> > Thank you for addressing my concerns, I am glad to hear you commit to re-emphasizing adherence to fair data use in the intro.  As this was my main concern I will now increase the score.

---

> > > ### Author Response · Authors · 2024-08-09
> > > **Author Response to Reviewer wVka**
> > >
> > > We thank the reviewer for their positive comments and the corresponding score increase. We further welcome any questions during the discussion period.

---

### Author Rebuttal · Authors · 2024-08-06

We thank the reviewers for their valuable time and effort in providing this review. We are encouraged and we appreciate the reviewer mentioning the simplicity, efficiency, and the strong performance of our approach. Below we provide some "global" comments that address questions shared by multiple reviewers.

**1. Does a goldfish model paraphrase its training data, even if it does not produce exact/verbatim copies?**

In Figure 1, we added Rouge1, Rouge2, and BERTScore [1], which represent unigram overlap, bigram overlap, and BERT embedding-based scores (a higher score suggests a closer semantic similarity to the ground truth), respectively. The goldfish model gets an embedding-based BERTScore of 75%, increased from the non-trained Control at 58%, and lesser than training with a standard loss at 97%. We also see a similar trend for n-gram-based Rouge scores indicating that goldfish models do generate paraphrases of training data, if not exact verbatim reproduction which is at 0% (same as Control and 85% for standard loss). This implies that the goldfish loss, as intended, deters the model from reproducing exact training samples during the inference phase. However, it still retains and learns knowledge from these samples, resulting in generated text that is semantically similar to the training data without being identical.

**2. Impact of goldfish-loss on downstream performance**
  - In Figure 2 (left), we plot the benchmark performance for control, goldfish, and standard models. We observe only marginal changes in performance between the 3 models, which is what we might expect from the original 1B TinyLLaMA work [2].
  - However, in Figure 5 of paper, we note that goldfish model have higher validation loss in comparison to standard models. We hypothesize that this validation loss gap is because of a lesser number of tokens _supervised_, i.e. optimized upon, during training.
  - To better understand this issue using a more sensitive experiment, we pre-trained an LLM on 300B tokens.  In Figure 2 (right), we run goldfish loss with proportionally increased _supervised_ token count by (i) increasing batch size in one experiment and (ii) training for more steps in another. Both configurations alleviate the validation loss gap and achieve final validation loss nearly identical to standard loss.  This shows that the performance loss of goldfish models is due to training on fewer tokens, and one can make up for this gap by training for longer.
  - In summary, goldfish loss observe a validation-loss gap in comparison to standard loss indicating some loss of performance. This suggests that for larger pretraining runs (>1B for >100B tokens, beyond our academic lab budget) and under identical settings, we would observe a decrease in benchmark scores for goldfish loss in comparison standard loss. From our new experiments, we also show that this can be alleviated by matching standard loss on number of _supervised_ tokens by either of above mentioned configurations.

---
[1] Tianyi Zhang, Varsha Kishore, Felix Wu, Kilian Q. Weinberger, & Yoav Artzi. (2020). BERTScore: Evaluating Text Generation with BERT.

[2] Peiyuan Zhang, Guangtao Zeng, Tianduo Wang, & Wei Lu. (2024). TinyLlama: An Open-Source Small Language Model.

---

### Decision · Program_Chairs · 2024-09-25

**Decision:**

Accept (poster)

**Comment:**

This paper proposes a simple and clever method to prevent language models from memorizing their output verbatim, based on modifying the next-token prediction loss such that it only applies to a random subset of the tokens. The approach seems effective and, especially based on follow-up experiments presented during the discussion, it doesn't seem to impact model performance too strongly.

The reviewers had concerns about the downstream benchmarks that were used, but these were at least partially addressed during the discussion.

There are two issues in terms of the general motivation for the paper that I think should be clearly addressed in the revision. First, the authors should be clear that, while the model might address the scenario in which it is legal to train on a certain text, as long as the text cannot be re-generated verbatim, it might still be the case that a certain text should not be used in model training. In this perspective, actually, the proposed model is legally problematic, as it allows model trainers to use texts they have no permission to use, while lowering the risk that they will be "discovered".

Second, and more "philosophically" (but I'd expect this to also have legal consequences), the authors should be clear that they aim to avoid verbatim reproduction of a text, while at the same time still allowing the model to generate a paraphrase of the text. It is not clear, to me and to a few reviewers, that this no longer counts as having memorized the text. It would be good to include a discussion of this point in the revision.